# *Toxoplasma gondii* GRA9 Regulates the Activation of NLRP3 Inflammasome to Exert Anti-Septic Effects in Mice

**DOI:** 10.3390/ijms21228437

**Published:** 2020-11-10

**Authors:** Jae-Sung Kim, Seok-Jun Mun, Euni Cho, Donggyu Kim, Wooic Son, Hye-In Jeon, Hyo Keun Kim, Kiseok Jang, Chul-Su Yang

**Affiliations:** 1Institute of Natural Science & Technology, Hanyang University, Ansan 15588, Korea; sung901017@naver.com; 2Department of Bionano Technology, Hanyang University, Seoul 04673, Korea; moon07101@hanyang.ac.kr (S.-J.M.); eunicho@hanyang.ac.kr (E.C.); 3Department of Molecular and Life Science, Hanyang University, Ansan 15588, Korea; ehdrb420@gmail.com (D.K.); wooic94@gmail.com (W.S.); jhi1007@naver.com (H.-I.J.); wow1560@naver.com (H.K.K.); 4Department of Pathology, Hanyang University College of Medicine, Seoul 04673, Korea; kiseok.jang@gmail.com

**Keywords:** *Toxoplasma gondii* GRA9, NLRP3, macrophages polarization, sepsis

## Abstract

Dense granule proteins (GRAs) are essential components in *Toxoplasma gondii*, which are suggested to be promising serodiagnostic markers in toxoplasmosis. In this study, we investigated the function of GRA9 in host response and the associated regulatory mechanism, which were unknown. We found that GRA9 interacts with NLR family pyrin domain containing 3 (NLRP3) involved in inflammation by forming the NLRP3 inflammasome. The C-terminal of GRA9 (GRA9C) is essential for GRA9–NLRP3 interaction by disrupting the NLRP3 inflammasome through blocking the binding of apoptotic speck-containing (ASC)-NLRP3. Notably, Q200 of GRA9C is essential for the interaction of NLRP3 and blocking the conjugation of ASC. Recombinant GRA9C (rGRA9C) showed an anti-inflammatory effect and the elimination of bacteria by converting M1 to M2 macrophages. In vivo, rGRA9C increased the anti-inflammatory and bactericidal effects and subsequent anti-septic activity in CLP- and *E. coli-* or *P. aeruginosa*-induced sepsis model mice by increasing M2 polarization. Taken together, our findings defined a role of *T. gondii* GRA9 associated with NLRP3 in host macrophages, suggesting its potential as a new candidate therapeutic agent for sepsis.

## 1. Introduction

*Toxoplasma gondii* (*T. gondii*) is an infectious intracellular parasite that is one of the most widespread zoonoses and causes toxoplasmosis. It has 23 excretory/secretory dense granule proteins (GRAs) [1,2]. Many studies have shown that GRA proteins actively regulate host gene expression through disturbing host cell transcriptional systems [3,4,5]. GRA9, one of the GRAs, is recognized as being essential for the tachyzoite and bradyzoite stages, which are stages at which *T. gondii* is infectious to its hosts. It is also suggested as a candidate for a vaccine for toxoplasmosis caused by *T. gondii* infection [6,7]. However, the role of *T. gondii* GRA9 in the host and how it regulates host immune system-related factors are unknown.

NLR family pyrin domain containing 3 (NLRP3) is a member of the leucine-rich repeat-containing receptors, which are a group of pattern recognition receptors that recognize the pathogen-associated molecular patterns and damage-associated molecular patterns derived from pathogens, damaged tissues, and cells [8]. In inflammation, NLRP3 constructs the NLRP3 inflammasome, which is assembled with adapter proteins termed apoptotic speck-containing protein (ASC) and the effector protein pro-caspase-1. The NLRP3 inflammasome leads to proximity-induced caspase-1 autoproteolysis, inducing the maturation of interleukin-1β (IL-1β) and IL-18 to ensure the secretion of inflammatory cytokines [9,10]. In addition, active caspase-1 induces the cleavage of gasdermin D, causing pro-inflammatory programmed cell death called pyroptosis [11]. NLRP3 signaling has emerged as an important signaling pathway of the innate immune system and the processing of inflammation. In addition, a variety of inflammatory diseases, including Parkinson’s disease, Alzheimer’s disease, inflammatory bowel disease, and atherosclerosis, are associated with the NLRP3 inflammasome [12,13,14]. In this study, GRA9 is shown to be a binding partner of NLRP3, but there have been limited studies of the role and mechanism of NLRP3 inflammasome activation with *T. gondii* GRA9.

Sepsis is defined as a life-threatening organ dysfunction caused by an uncontrolled host response to infection. Every year, sepsis is a major issue in public health because of its high incidence and mortality rates worldwide [15]. The early stage of sepsis is related to hyper-inflammation, in which there is an increase in the release of inflammatory cytokines and chemokines from immune cells, termed “systemic inflammatory response syndrome” [16]. For the treatment of sepsis, adequate antibiotics are required for the control of bacterial infection. Antibiotics are selected depending on the species of bacteria and the condition of the patient [17]. Regulating the host immune response is also a strategy for treating sepsis. Many pattern recognition receptors are known to be involved in the activation of inflammation in sepsis [18]. Some studies have shown that NLRP3 is a key factor involved in inflammation in sepsis. The activation of the NLRP3 inflammasome during sepsis is related to mitochondrial dysfunction and organ injury [19,20,21].

In this study, we demonstrated that GRA9 interacted with NLRP3 and inhibited the formation of the NLRP3 inflammasome through the blockade of binding of ASC to NLRP3 in mitochondria. In particular, the C-terminal of GRA9 was essential for association with NLRP3, related to anti-inflammatory and bactericidal effects through M2 polarization, and showed anti-septic effects in vivo. Therefore, *T. gondii* GRA9 is a potential candidate for a therapeutic agent acting via interaction with NLRP3 for sepsis.

## 2. Results

### 2.1. GRA9 Interacts with NLRP3

To clarify the function of *T. gondii* GRA9 in host immune responses in macrophages, we investigated whether it interacts with components involved with inflammation. GRA9 complexes were subjected to co-immunoprecipitation with THP-1 cells containing a vector or Flag-GRA9. The purified GRA9 complexes were identified by mass spectrometry analysis, where they were shown to contain NLRP3 (118K) and metallothionein 2A (MT2A, 6K) (Figure 1A). NLRP3 is a cytosolic receptor related to the activation of immune response in macrophages [9]. In previous studies, *T. gondii* GRA proteins were shown to interact with host proteins associated with immune cells and modulated the host immune system [22,23,24]. However, the interaction between *T. gondii* GRAs and NLRP3 is unclear. To investigate the relationship between NLRP3 and GRA9, we primed THP-1 cells by treatment with LPS and activated them by treatment of ATP or nigericin. In GRA9-expressing THP-1 cells, endogenous co-immunoprecipitation showed that GRA9 interacted strongly but temporarily (from 30 to 60 min) with endogenous NLRP3 after priming with LPS and stimulation with ATP or nigericin (Figure 1B). Taking these findings together, we identify that GRA9 interacts with NLRP3 in macrophages.

### 2.2. GRA9 Interacts with NLRP3 in Mitochondria by Inhibiting the Assembly of NLRP3 Inflammasome by Blocking the Binding of ASC

Mitochondria are organelles related to induction of the recruitment of NLRP3 to construct the NLRP3 inflammasome in mitochondria-related membrane (MAM) through the production of mitochondrial reactive oxygen species [25]. Since we found that GRA9 is a binding partner of NLRP3, we examined its localization in macrophages. We stimulated THP-1 cells with LPS and activated them by treating them with ATP, followed by the subcellular fractionation of organelles. In GRA9-expressing THP-1 cells, we identified GRA9 and NLRP3 as being mainly localized in MAM and mitochondria, through immunoblotting and fluorescence imaging in each subcellular fraction (Figure 2A,B). To determine whether GRA9 interacted with NLRP3 in mitochondria, we immunoprecipitated the NLRP3 in the cytosolic or mitochondrial fraction in vector- or GRA9-expressing THP-1 cells. Notably, not only did GRA9 interact with NLRP3 but it also blocked the binding of ASC that interacted with NLRP3 in order to generate NLRP3 inflammasome in the mitochondria, not the cytosol (Figure 2C). Consistent with the findings in Figure 2C, fluorescence imaging showed that NLRP3 and GRA9 were co-localized in THP-1 cells (Figure 2D). In addition, we transfected V5-GRA9, Flag-NLRP3, and AU1-ASC plasmids and immunoprecipitated Flag or AU1. As shown in previous data, GRA9 interacted with NLRP3 and inhibited the association of ASC in a manner dependent on the dose of GRA9 (Figure 2E). Taking these findings together, GRA9 acts as a binding partner of NLRP3 by competitively inhibiting the binding of ASC in mitochondria.

### 2.3. C-Terminal of GRA9 Decreases the Activation of NLRP3 Inflammasome via Interaction with NLRP3

To investigate the domain of interaction between GRA9 and NLRP3 in macrophages, we immunoprecipitated the GRA9 and NLRP3 WT or mutants. GRA9 contains a signal sequence, an N-terminal domain (amino acids (aa) 18–171), and a C-terminal domain (aa 195–288) (Figure 3A). In 293T cells, domain mapping using various mammalian fusions and truncated mutants of GST-GRA9 or Flag-NLRP3 with wild-type Flag-NLRP or V5-GRA9 indicated that the C-terminal domain of GRA9 had minimal binding affinity with NLRP3, and the LRR domain (aa 742–1036) of NLRP3-bound GRA9 as strongly as the wild-type NLRP3. Additionally, a Q200L mutant in the C-terminal of GRA9 had no effect on the interaction between GRA9 and NLRP3. This demonstrated that Q200 in the C-terminal of GRA9 is a vital amino acid for the interaction with NLRP3 (Figure 3A). Construction of the NLRP3 inflammasome with ASC and pro-caspase 1 is induced with activation of the immune response in the host. Caspase 1 is a cleaved form of pro-caspase 1, which is important for the cleavage of IL-1β and IL-18 for their secretion from macrophages [9]. To examine the role of GRA9 in macrophages, we performed immunoblotting and measured the cytokines related to inflammation. In THP-1 cells expressing the C-terminal of GRA9 (GRA9C), the cleavage and secretion of caspase 1, IL-1β, and IL-18 were decreased after stimulation with LPS and activation by ATP or DSS in supernatant, compared with the levels for vector- or GRA9^Q200L^-expressing THP-1 cells (Figure 3B,C). Interestingly, the levels of TNF-α, IL-6, and IL-12 showed no significant differences in GRA9C-expressing THP-1 cells compared with those in vector- and GRA9^Q200L^-expressing ones. This showed that the interaction of GRA9C with NLRP3 is only related to the activation stage of NLRP3 inflammasome (Figure 3D). Taking these findings together, the C-terminal of GRA9, especially Q200, is essential for interaction with NLRP3 and blocks NLRP3-induced inflammation by inhibiting the assembly of the NLRP3 inflammasome.

### 2.4. rGRA9C Attenuates the NLRP3-Induced Inflammation by Interacting NLRP3

To identify the function of GRA9C under physiological conditions, we produced bacterially purified His-tagged rGRA9C or rGRA9C^Q200L^ proteins and confirmed their identity using SDS-polyacrylamide gel electrophoresis and immunoblotting (Figure 4A). No significant differences compared with the vector controls were observed regarding rGRA9C- and rGRA9C^Q200L^-induced cytotoxicity in macrophages (Figure 4B). We also performed rGRA9C treatment in bone marrow-derived macrophages (BMDMs) and observed the localization through confocal imaging. Consistent with the previous results, rGRA9C was co-localized with NLRP3, but rGRA9C^Q200L^ was not (Figure 4C). To examine whether rGRA9C regulates the NLRP3-induced inflammation in macrophages, we performed LPS and ATP treatments for priming and activation of the NLRP3 inflammasome and performed rVector, rGRA9C, or rGRA9C^Q200L^ treatment in BMDMs. Consistent with the results in Figure 3, rGRA9C was associated with NLRP3 by blocking the ASC binding with NLRP3 in a dose-dependent manner, but rGRA9C^Q200L^ exhibited no binding with NLRP3. In addition, the secretion and cleavage of IL-1β, IL-18, and caspase 1 were inhibited by rGRA9C in BMDMs (Figure 4D,E). Similarly, we observed that the levels of IL-1β and IL-18 were decreased by treatment with rGRA9C, but not by that with rGRA9C^Q200L^, in BMDMs stimulated with LPS and ATP or DSS (Figure 4F). Taking these findings together, rGRA9C inhibits NLRP3-induced inflammation via interaction with NLRP3 in macrophages.

### 2.5. rGRA9C Enhances Anti-Inflammatory and Bactericidal Effects via M1 to M2 Polarization

Macrophages are essential for innate immune responses and play important roles including in phagocytosis, activation of inflammation, and wound repair [26]. To determine the effect of GRA9C related to macrophage function, we performed LPS treatment of BMDMs with rVehicle, rGRA9C, or rGRA9C^Q200L^ at various doses. We measured the levels of pro- and anti-inflammatory cytokines by ELISA. Discordantly with Figure 3, rGRA9C partially decreased the levels of pro-inflammatory cytokines (TNF-α and IL-6) and increased those of anti-inflammatory cytokines (IL-10 and TGF-β) (Figure 5A). Since the roles of macrophages are distinct depending on the presence of M1 or M2 polarization, we assessed whether rGRA9C is related to macrophage polarization. rGRA9C showed increases of M2 markers (CD163 and Arg1) along with decreases of M1 markers (CD86 and iNOS), but this was not the case for rGRA9C^Q200L^ (Figure 5B). In addition, rGRA9C increased the expression of scavenger receptor A (SR-A) and Fc receptor, which are related to phagocytosis [27,28]. Interestingly, the expression of TLR4 was enhanced, while that of TLR2 and TLR6 was not (Figure 5C). These results demonstrated that rGRA9C is related to regulation of the roles of macrophages by converting M1 to M2 macrophages. Additionally, we examined whether GRA9C mediated the bactericidal effect in macrophages through infected bacteria. *E. coli* and *P. aeruginosa* were eliminated significantly in rGRA9C-treated BMDMs in a dose-dependent manner, but this was not the case for rGRA9C^Q200L^-treated ones (Figure 5D,E). These results showed not only that rGRA9C regulated the activation of the NLRP3 inflammasome through interacting with NLRP3, but also that it regulated the functions of macrophages. Taking these findings together, rGRA9C increases the anti-inflammatory and bactericidal effects through increasing M2 polarization and phagocytosis.

### 2.6. rGRA9C Protects Mice from CLP and Bacteria-Induced Sepsis

We aimed to determine whether rGRA9C protects mice from septic shock due to polymicrobial peritonitis using a cecal ligation and puncture (CLP) model of polymicrobial infection or bacterial infection, which triggers systemic inflammatory response syndrome and is typically fatal in humans [29,30]. First, we tested the protective effect of rGRA9C and mutant against CLP-induced death in mice. We induced a sepsis model via CLP and applied control, rGRA9C, or rGRA9C^Q200L^ treatment via intraperitoneal injection. Notably, rGRA9C prevented and protected the mice from CLP-induced septic shock in a dose-dependent manner compared with the control, but this was not the case for GRA9C^Q200L^ (Figure 6A). Furthermore, the levels of cytokines in serum were regulated in rGRA9C-treated mice. Pro-inflammatory cytokines including TNF-α, IL-6, IL-1β, IL-18, and IL-12p40 were decreased, while anti-inflammatory cytokines including IL-10 and TGF-β were increased in rGRA9C-treated mice (Figure 6B). We examined the bacterial clearance and found that treating CLP mice with rGRA9C also increased the bactericidal effect in peritoneal fluid and blood (Figure 6C). These results were accompanied by reduced infiltration of immune cells and decreased damage to the lung, liver, and spleen, as revealed by hematoxylin and eosin staining (Figure 6D). In rGRA9C-treated mouse spleen, rGRA9C was associated with NLRP3 by preventing the binding of ASC, in accordance with the THP-1 or BMDM results (Figure 6E). Additionally, we assessed whether GRA9C was specifically related to macrophages or whether other immune cells were also involved. Intriguingly, the treatment of rGRA9C in CLP-induced mice significantly increased the number of macrophages, but other immune cells showed no significant difference (Appendix A). Through the expression of M1 or M2 markers in mouse spleen, polarization of M1 to M2 macrophages was increased in CLP-induced mice with GRA9C (Figure 6F).

To evaluate the effect of GRA9C in bacterial-induced sepsis, we infected the mice with *Escherichia coli (E. coli) or Pseudomonas aeruginosa (P. aeruginosa)* through intravenous or intraperitoneal injection, respectively, and performed treatment with the rGRA9C or mutant. rGRA9C significantly increased the survival rate of mice by about 60% compared with the findings for control or rGRA9C^Q200L^-treated mice (Figure 7A). In LB broth, rGRA9C had no bactericidal effect, which showed that rGRA9C needs to interact with NLRP3 to exert its anti-microbial function (Figure 7B). Bacterial colony-forming units in blood and peritoneal fluid were decreased in *E. coli*- or *P. aeruginosa*-infected mice with rGRA9C (Figure 7C). Furthermore, consistent with Figure 6F, the polarization of M1 to M2 macrophages was also increased in mice treated with rGRA9C, but not rGRA9C^Q200L^ (Figure 7D). Taking these findings together, rGRA9C increases the anti-inflammatory and bactericidal effects through interacting with NLRP3 in CLP-induced and bacterial-induced sepsis model mice.

## 3. Discussion

This study described a novel anti-sepsis therapeutic candidate based on *T. gondii* GRA9-induced anti-inflammatory and bactericidal effects. The major findings of this study are as follows: (1) GRA9 interacts with NLRP3 by blocking the binding of ASC; (2) GRA9 co-localizes with NLRP3 in mitochondria to regulate the activation of NLRP3 inflammasome; (3) the C-terminal of *T. gondii* GRA9 is essential for interaction with NLRP3 and the Q200 residue is required for this interaction; (4) the C-terminal of GRA9C inhibits the activation of NLRP3 inflammasome with decreasing cleavage of pro-IL-1β and pro-IL-18; (5) rGRA9C shows not only inhibition of activation of NLRP3 inflammasome-induced inflammation but also anti-inflammatory and bactericidal effects by upregulating the polarization of M1 to M2 macrophages and phagocytosis in BMDMs; and (6) rGRA9C enhances the anti-septic effect by interacting with NLRP3 in CLP- and bacteria-induced model mice.

*T. gondii* GRA proteins are secreted during and after invasion into the parasitophorous vacuole. *T. gondii* GRA proteins are essential for successful infection of *T. gondii* by facilitating the regulation of host factors that interact with host proteins or transcription factors. Using a yeast two-hybrid technique in a HeLa cell cDNA library, there are host proteins which are interacted with *T. gondii* GRA9, including filamin B beta (FLNB, NM_001457), metallothionein 2A (MT2A, NM_005953), and processing of precursor 7 ribonuclease P subunit (POP7, NM_005837) [31]. Our results are in accordance with those of a previous study; we also suggested NLRP3 as a novel binding partner to GRA9 by immunoprecipitation in THP-1 cells. We previously found that ASC, a component of the NLRP3 inflammasome, interacted with *T. gondii* GRA7 by enhancing protective defense of the host against bacteria [32]. However, the interaction between *T. gondii* proteins and NLRP3 is unclear. Thus, GRA9-induced regulation of the NLRP3 inflammasome may contribute to studying how *T. gondii* GRA proteins are involved in activation of the NLRP3 inflammasome.

In our previous studies, we discovered the roles of other *T. gondii* GRA proteins, GRA7 and GRA8, which regulate host signaling by interacting with host proteins. GRA7 was shown to enhance host immunity by activating TRAF6 [23]. We also found that GRA7 interacted with host proteins ASC and PLD1 via PKCα. The interaction of GRA7 with binding partners induced anti-bacterial effects in *M. tuberculosis* infection [32]. GRA8 enhanced metabolism via the ATP5A1–SIRT3 pathway and showed a therapeutic effect in sepsis [33]. In addition, a GRA8-derived peptide, related to the ATP5A1–SIRT3 pathway, was reported to show an anti-tumor effect in colorectal cancer [34]. These studies showed that *T. gondii* GRA7 and GRA8 proteins regulate host inflammation and signaling by associating with host proteins and could be therapeutic candidates in other diseases. Given the role of GRA9, it is also a potential candidate for treating sepsis by alleviating inflammation and killing the causative bacteria.

The NLRP3 inflammasome is a supramolecular complex sensor for activating inflammation and pyroptosis by releasing IL-1β and IL-18. To activate the NLRP3 inflammasome, two stages are required: The first stage is priming, whereby the expression of Nlrp3 and Il1b genes is upregulated, followed by construction of the NLRP3 inflammasome. The second step is activation, triggering the assembly of the NLRP3 inflammasome complex and the activation of caspase 1 [9]. We show the different results regarding the levels of TNF-α and IL-6 in Figure 3D and Figure 5A. We speculate that the endogenous expression of GRA9 only regulated the activation step in the NLRP3 inflammasome, but rGRA9C mediated both steps. These results showed that rGRA9C should be related to cellular membrane receptors and NF-κB signaling. In further study, we need to study the interaction between rGRA9C and membrane receptor signaling, such as that of TLRs.

NLRP3 is related to a variety of cellular processes, including K^+^ efflux, Ca^2+^ uptake, lysosomal disruption, metabolism, and mitochondrial dysfunction [35]. In LPS treatment, cellular metabolism switches from oxidative phosphorylation to glycolysis, contributing to activation of the NLRP3 inflammasome. Furthermore, the accumulation of succinate increases the level of IL-1β in a manner dependent on HIF-1α induction [36,37]. Mitochondrial dysfunction is related to NLRP3 activation through multiple mechanisms. A danger signal or metabolic stress initiates mitochondrial ROS and DNA, which is the ligand to trigger NLRP3 activation and cause apoptosis [38,39]. The mitochondrial lipid cardiolipin was also reported to be a ligand for activating the NLRP3 inflammasome in a linezolid-induced model [40]. Since activation of the NLRP3 inflammasome is essential for the host immune response, NLRP3 is associated with many diseases, including Alzheimer’s disease, atherosclerosis, gout, and inflammatory bowel disease. In most of these diseases, NLRP3 activation is dysregulated, causing chronic inflammation and the accumulation of danger signals [35]. In this study, we observed the interaction of GRA9 with NLRP3 in mitochondria. We suggested that the translocation of GRA9 to the mitochondria is essential for regulation of the NLRP3 inflammasome. However, the mechanism behind the localization of GRA9 to mitochondria and whether GRA9 affects mitochondrial functions are unclear.

Macrophages are essential for host immune responses related to both innate and adaptive immunity. They are involved in numerous cellular functions including inflammation, wound healing, and anti-bacterial and anti-inflammatory effects. The polarization of macrophages is essential for the selection of roles in macrophages [41,42,43]. The major phenotypes of macrophages are classically activated macrophages (M1 macrophages) and alternatively activated macrophages (M2 macrophages). M1 macrophages are stimulated by TLR agonist or IFN-γ. They produce pro-inflammatory cytokines or chemokines for enhancing inflammation, killing bacteria, and activating T cells. Meanwhile, M2 macrophages are polarized by anti-inflammatory factors, such as IL-4, IL-10, and IL-13. These macrophages are related to the resolution of inflammation and recovery of tissue. M2 macrophages increase the production of anti-inflammatory cytokines or chemokines, such as IL-4, IL-10, and TGF-β [44,45]. The functions of both M1 and M2 macrophages are vital for the homeostasis of immunity and an adequate balance between these macrophages is related to defense against many infectious or inflammatory diseases [46]. In our study, we showed that rGRA9C regulated the activation of NLRP3 and the shift of polarization to M2 macrophages from M1 macrophages. We assumed that this shift of macrophage polarization is involved in activation of the NLRP3 inflammasome. In a previous study, the NLRP3 inflammasome was shown to regulate M2 polarization via the upregulation of IL-4 in asthma [47]. Furthermore, another study showed that metformin, an inhibitor of glycolysis, induced M2 macrophage polarization through mediating the AMPK–mTOR–NLRP3 inflammasome signaling pathway [48]. Taking these findings together, further study is required to understand how GRA9 regulates the macrophage polarization via interacting with NLRP3.

Sepsis is a well-known systemic inflammatory disease causing life-threatening organ dysfunction. In the early phase of sepsis, hyper-inflammation is induced by activation of the immune cells and causes organ dysfunction and death. In the late phase of sepsis, immunosuppression is initiated by paralyzing the immune system, causing an immune imbalance [17]. The regulation of NLRP3 inflammasome activation is important for both stages of sepsis. In a CLP-induced sepsis model, NLRP3-deficient mice showed high resolution of inflammation and a high survival rate compared with wild-type mice [49]. However, NLRP3 activation was shown to be impaired by the P2 × 7 receptor in monocytes from a patient with an immunocompromised status caused by mitochondrial dysfunction and the inhibition of NLRP3 activation by HIF-1α [50]. However, the mechanism of regulation of the NLRP3 inflammasome in sepsis has remained unclear.

Taking the findings presented here together, we provided evidence that GRA9 plays a novel role in regulating NLRP3 inflammasome activation and macrophage polarization in the host. We suggested that GRA9-mediated regulation could be a strategy for treating sepsis. However, further studies are required to understand the detailed mechanism behind the role of *T. gondii* GRA9 in the host in order to use it as a therapeutic candidate in other inflammatory or infectious diseases.

## 4. Materials and Methods

### 4.1. Mice and Cell Culture

Wild-type C57BL/6 mice were purchased from Samtako Bio Korea (Gyeonggi-do, Korea). Primary bone marrow–derived macrophages (BMDMs) were isolated from six-week-old C57BL/6 mice and cultured in DMEM for 3–5 days in the presence of M-CSF (R&D Systems, 416-ML), as described previously [32]. HEK293T (ATCC-11268; American Type Culture Collection) or THP-1 (ATCC-TIB-202) cells were maintained in DMEM or RPMI1640 (Gibco, NY, USA) containing 10% FBS (Gibco, NY, USA), sodium pyruvate, nonessential amino acids, penicillin G (100 IU/mL), and streptomycin (100 μg/mL). Transient transfections were performed using calcium phosphate (Clontech, Mountain View, CA, USA) in 293T, according to the manufacturer’s instructions. THP-1 stable cell lines were generated by transfections were performed using Lipofectamine 3000 (Invitrogen, Waltham, MA, USA) and then a standard selection protocol with 400–800 μg/mL of G418.

### 4.2. Recombinant Protein

To obtain *T. gondii* ME49 strain-derived recombinant GRA9 (GenBank accession no. XP_002367395.1) protein, GRA9C amino acid residues (195 to 288) and GRA9C^Q200L^ were cloned with an N-terminal 6xHis tag into the pRSFDuet-1 Vector (Novagen, NJ, USA) and induced, harvested, and purified from *Escherichia coli* expression strain BL21(DE3) pLysS as described previously [34], in accordance with the standard protocols recommended by Novagen. rGRA9 was dialyzed with permeable cellulose membrane and tested for lipopolysaccharide contamination with a *Limulus* amebocyte lysate assay (BioWhittaker, Walkersville, MD, USA) and contained <20 pg/mL at the concentrations of rGRA9 and mutant protein used in the experiments described here.

### 4.3. Reagents and Antibodies

LPS (L3024), ATP (A1852), Nigericin (N7143), and DSS (D8906) were purchased from Sigma-Aldrich (St. Louis, MO, USA). Abs specific for Flag (D-8), GST(B-14), V5 (E10), NLRP3 (H-66), ASC (N-15), TXNIP (D-2), Tubulin (5F131), Calnexin (AF18), FACL4 (N-18), caspase-1 (14F468, M-20), and Actin (I-19) were purchased from Santa Cruz Biotechnology (Dallas, Texas, USA). The antibody to calreticulin (D3E6) and IL-18 (D2F3B) was from Cell Signaling Technology (Danvers, MA, USA), IL-1β (AF-401-NA) was from R&D Systems and NLRP3 (AG-20B-0014) was from Adipogen (San Diego, CA, USA). The antibodies to COX IV (ab16056) and AU1 (ab3401) were purchased from Abcam (Cambridge, UK).

### 4.4. Plasmid Construction

The plasmid encoding full-length of the NLRP3, ASC, and NLRP3 mutant plasmids were previously described [23]. Plasmids encoding different regions of GRA9 (1-318, 18-171, 195-288, Q200L) were generated by PCR amplification from full-length GRA9 cDNA and subcloning into a pEBG derivative encoding an N-terminal GST epitope tag between the *BamHI* and *NotI* sites. All constructs for transient and stable expression in mammalian cells were derived from the pEBG-GST mammalian fusion vector and the pEF-IRES-Puro expression vector. All constructs were sequenced using an ABI PRISM 377 automatic DNA sequencer (Thermofisher, Waltham, MA, USA) to verify 100% correspondence with the original sequence.

### 4.5. Enzyme-Linked Immunosorbent Assay

Cell culture supernatants and mice sera were analyzed for cytokine content using the BD OptEIA ELISA set (BD Pharmingen, San diego, CA, USA) for the detection of TNF-α, IL-6, IL-1β, IL-18, IL-12p40, IL-10, and TGF-β. All assays were performed as recommended by the manufacturer.

### 4.6. CLP-Induced Sepsis and Bacteria Counts

Cecal ligation and puncture (CLP) were performed using 6-week-old C57BL/6 female mice (Samtako Bio, Gyeonggi-do, Korea), as described previously [29,33]. For CLP, mice were anesthetized with pentothal sodium (50 mg/kg, *i.p.*), and a small abdominal midline incision was made to expose the cecum. The cecum was then ligated below the ileocecal valve, punctured twice through both surfaces, using a 22-gauge needle, and the abdomen was closed. The survival rate was monitored daily for 10 days. The mice were resuscitated by intraperitoneal injection of PBS, analgesic (1.5 mg/kg nalbuphine; Sigma-Aldrich, St. louis, Missouri, USA), and an antibiotic cocktail containing ceftriaxone (25 mg/kg; Sigma-Aldrich, St. louis, Missouri, USA), and metronidazole (12.5 mg/kg; Sigma-Aldrich, St. louis, Missouri, USA) in 100 μL PBS at 12 h and 24 h after CLP onset [51,52]. For experiments aimed to isolate blood and organ samples, sham-operated mice of which the cecum was exposed but not ligated or punctured were used and are indicated as sham at the time of the surgery.

For bacteria count, blood was collected by cardiac puncture or peritoneal lavage fluids from mice at indicated time after CLP. After performing serial dilution of blood, 5 μL of each dilution was plated on blood agar plates. Bacteria were counted after incubation at 37 °C for 24 h and calculated as counting colony-forming units per blood or whole peritoneal lavage.

All animals were maintained in a pathogen-free environment. All animal experimental procedures were reviewed and approved by the Institutional Animal Care and Use Committee of Hanyang University (protocol 2018-0086, Approval date: 8 June 2018). CLP model that post-CLP analgesia, fluid support and antibiotics is consistent with international guidelines, defined as the ‘‘Minimum Quality Threshold in Pre-Clinical Sepsis Studies’’ for the sepsis mouse model, to enhance translational relevance of the models [53,54].

### 4.7. GST Pulldown, Immunoblot, and Immunoprecipitation Analysis

GST pulldown, immunoprecipitation, and immunoblot assays were performed as previously described [32,33]. Details of the method are provided in the Appendix A.

### 4.8. Histology

For immunohistochemistry of tissue sections, mouse spleens, livers, and lungs were fixed in 10% formalin and embedded in paraffin. Paraffin sections (4 μm) were cut and stained with hematoxylin and eosin (H and E). Histopathologic score was established on the basis of the numbers and distribution of inflammatory cells and the severity of inflammation within the tissues [55,56] in which a board-certified pathologist independently scored each organ section without prior knowledge of the treatment groups. A histological score ranging from 0–4 was ascribed to each specimen.

### 4.9. Miscellaneous Procedures

Details of quantitative real-time polymerase chain reaction (PCR), protein purification and mass spectrometry, confocal fluorescence microscopy, cellular fractionation, MTT assay, and flow cytometry are provided in the Appendix A.

### 4.10. Statistical Analysis

All data were analyzed using Student’s *t*-test with Bonferroni adjustment for multiple comparisons and are presented as mean ± SD. Statistical analyses were conducted using the SPSS (Version 12.0) statistical software program (SPSS, Chicago, IL, USA). Differences were considered significant at *p* < 0.05. For survival, data were graphed and analyzed by the product limit method of Kaplan and Meier, using the log-rank (Mantele–Cox) test for comparisons using GraphPad Prism (version 5.0, La Jolla, CA, USA).

## Figures and Tables

**Figure 1 ijms-21-08437-f001:**
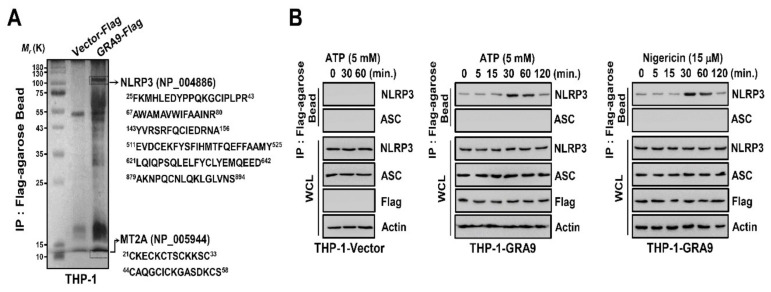
GRA9 binds with NLRP3. (**A**) Identification of NLRP3 by mass spectrometry analysis in THP-1 cell lysates expressed with GRA9 or vector. (**B**) Vector or GRA9 expressing THP-1 cells were primed with LPS (100 ng·mL^−1^) and stimulated with ATP or Nigericin (5 mM or 15 μM) for the indicated times, followed by immunoprecipitation (IP) with αFlag-agarose bead and immunoblotting (IB) with αNLRP3, αASC, αFlag, and αActin. The data are representative of four independent experiments with similar results (**A**,**B**). Full-length images of the blots presented in the Appendix A.

**Figure 2 ijms-21-08437-f002:**
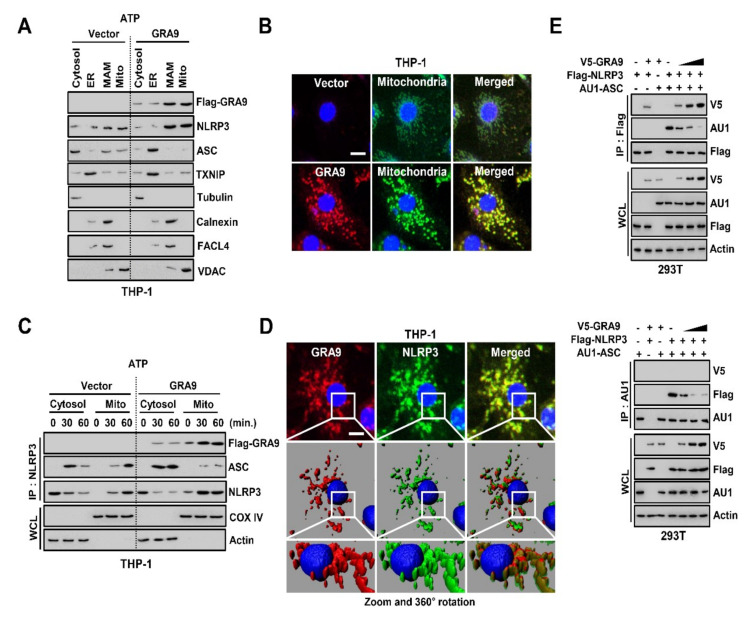
GRA9 associates with NLRP3 inhibiting the binding of ASC in mitochondria. (**A**) Subcellular fractionation of THP-1 cells stably expressing either Vector or GRA9 primed with LPS (100 ng mL^−1^) and stimulated with ATP (5 mM). Cytosolic, endoplasmic reticulum (ER), mitochondrial-associated membrane (MAM) and mitochondrial fractions were fractionated and analyzed for GRA9, NLRP3, and ASC by immunoblotting (IB). Purify of the fraction was assessed by blotting for TXNIP (thioredoxin-interacting protein; ER protein), tubulin (cytosolic protein), calnexin (MAM protein), FACL4 (acyl-CoA synthetase 4; MAM protein), and VDAC (voltage-dependent anion channel; mitochondrial protein). (**B**) Fluorescence confocal images in THP-1 cells expressing either Vector or rGRA9 were primed with LPS and stimulated with ATP and immunolabeled with αFlag (Alexa 568), MitotrackerDeep Green and DAPI. Scale bar, 10 μm. (**C**) Vector or GRA9 expressing THP-1 cells were primed LPS (100 ng mL^−1^) and stimulated with ATP (5 mM) for the indicated times, followed by fractionation. Fractionation of each THP-1 cells were immunoprecipitated by NLRP3 and analyzed by IB with αGRA9, αASC and αNLRP3. Whole cell lysates (WCLs) were used for IB with αCOX IV (mitochondrial protein), and αactin (cytosolic protein). (**D**) Fluorescence confocal images in THP-1 cells expressing either Vector or rGRA9 were primed with LPS and stimulated with ATP and immunolabeled with αFlag (Alexa 568), αNLRP3 (Alexa 488) and DAPI and analyzed in three-dimension. Scale bar, 10 μm. (**E**) At 24 h after transfection with V5-GRA9, Flag-NLRP3, and AU1-ASC, 293T cells were immunoprecipitated by Flag or AU1, followed IB with αV5, αFlag, and αAU1. Whole cell lysates (WCLs) were used for IB with αV5, αFlag, αAU1, and αactin. The data are representative of four independent experiments with similar results (**A**–**E**). Full-length images of the blots presented in the Appendix A.

**Figure 3 ijms-21-08437-f003:**
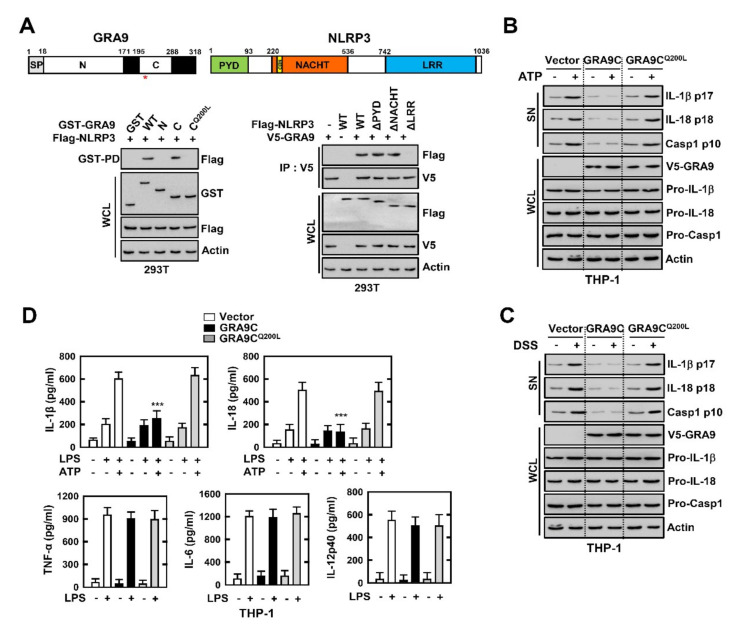
C-terminal of GRA9 attenuates the induction of NLRP3 inflammasome through binding with NLRP3. (**A**) Binding mapping. Schematic diagrams of the structures of GRA9 and NLRP3 (upper). At 48 h after transfection with mammalian glutathione S-transferase (GST) or GST-GRA9 and truncated mutant constructs together with Flag-NLRP3 or Flag-NLRP3 constructs together with V5-GRA, 293T cells were used for GST pull down or immunoprecipitated by αV5, followed by IB with αFlag, or αV5. Cell lysates (WCLs) were used for IB with αGST, αV5, αFlag, αV5, or αActin. (**B**,**C**) Vector, GRA9C or GRA9C^Q200L^ expressing THP-1 cells primed with LPS (100 ng mL^−1^) for 4 h and stimulated with ATP (5 mM) or DSS (3%) for 1 h or 18 h, followed by IB in supernatant (SN) with αIL-1β p17, αIL-18 p18 or αCasp1 p10 and WCL with αPro-IL-1β, αPro-IL-18, αPro-Casp1, or αActin. (**D**) Level of cytokines is analyzed by ELISA for IL-1β, IL-18, TNF-α, IL-6, and IL-12p40. The data are representative of four independent experiments with similar results (**A**–**D**). (**D**) Significant differences (*** *p* < 0.001) compared with rVector-expressing THP-1 cells. Full-length images of the blots presented in the Appendix A.

**Figure 4 ijms-21-08437-f004:**
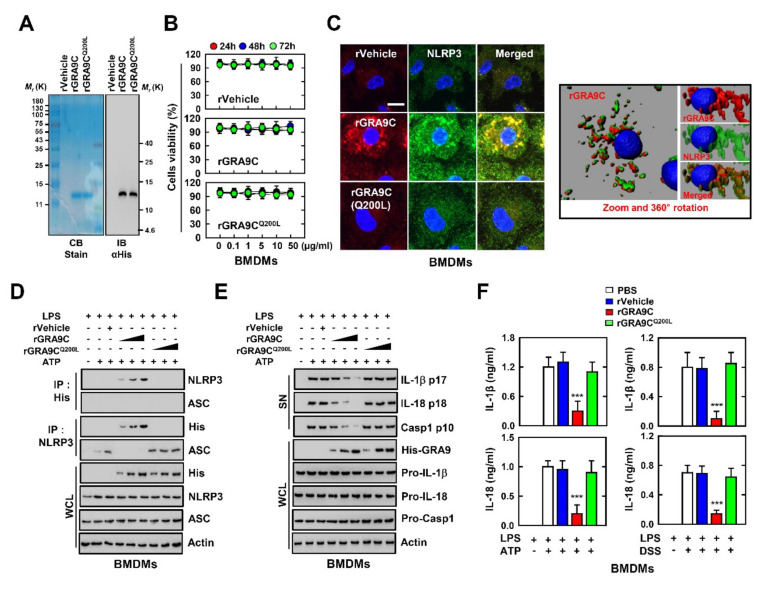
rGRA9C decreases the activation of NLRP3 inflammation by associating NLRP3. (**A**) Bacterially purified 6xHis-GRA9C and rGRA9C^Q200L^ were analyzed by Coomassie blue staining (left) or immunoblotting (IB) with αHis (right). (**B**) BMDMs were incubated with rGRA9C and rGRA9C^Q200L^ for the indicated times and concentrations and cell viability measured with MTT assay. (**C**) BMDMs were treated with rVehicle, rGRA9C, or rGRA9C^Q200L^, and immunolabelled with αFlag (Alexa 586), αNLRP3 (Alexa 488) and DAPI and analyzed in three dimensions. (**D**,**E**) BMDMs were primed with LPS (100 ng mL^−1^) and stimulated with ATP (5 mM) and rVehicle, rGRA9C or rGRA9C^Q200L^ at various concentrations (0.1, 1, or 10 μg/mL). Followed by IP with αHis or αNLRP3 and immunoblotted with αNLRP3, αASC or αHis (D) or IB with in supernatant (SN) with αIL-1β p17, αIL-18 p18, or αCasp1 p10 and WCL with αHis, αPro-IL-1β, αPro-IL-18, αPro-Casp1, or αActin (**E**). (**F**) BMDMs were primed with LPS (100 ng mL^−1^) and stimulated with ATP (5 mM) or DSS (3%) and rVehicle, rGRA9C, or rGRA9C^Q200L^. The level of cytokines is analyzed by ELISA for IL-1β, and IL-18. The data are representative of four independent experiments with similar results (**A**–**E**). (**F**) Significant differences (*** *p* < 0.001) compared with rVehicle-treated BMDMs. Full-length images of the blots presented in the Appendix A.

**Figure 5 ijms-21-08437-f005:**
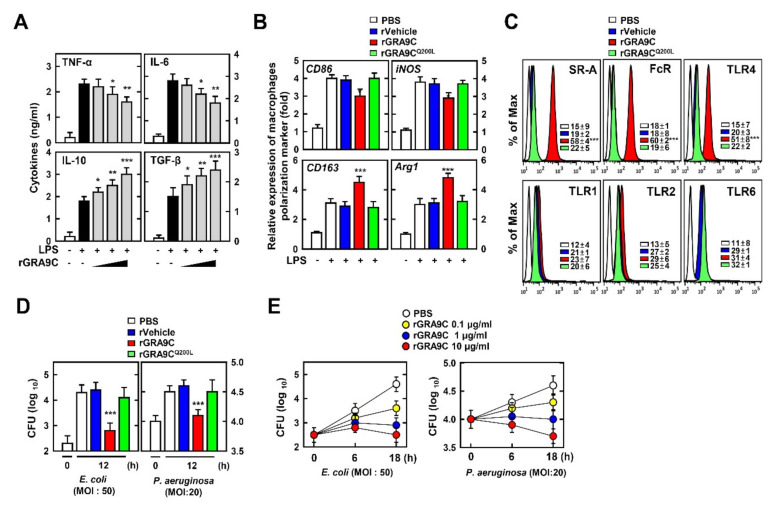
rGRA9C enhances anti-inflammation and bactericidal effect via M1 to M2 polarization. (**A**) BMDMs were primed with LPS (100 ng mL^−1^) and stimulated with rGRA9C at various concentrations (0.1, 1, or 10 μg/mL), followed by analysis of the level of cytokines is analyzed by ELISA for TNF-α, IL-6, IL-10 and TGF-β. (**B**) BMDMs were primed with LPS (100 ng mL^−1^) and stimulated with rVehicle, rGRA9C, or rGRA9C^Q200L^. Macrophage polarization markers in BMDM were measured by quantitative real-time PCR. (**C**) FACS analysis for SR-A, FcR, TLR4, TLR1, TLR2, and TLR6 from BMDMs. (**D**) BMDM were infected with *E. coli* (MOI = 50) or *P. aeruginosa* (MOI = 20) for 4 h and stimulated with rVehicle, rGRA9C, or rGRA9C^Q200L^ (1 μg/mL) for 12 h. BMDMs were lysed to determine intracellular bacterial loads. (**E**) BMDM were infected with *E. coli* (MOI = 50) or *P. aeruginosa* (MOI = 20) for 4 h and stimulated with PBS, or rGRA9C at various concentrations for indicated times. BMDMs are lysed to determine intracellular bacterial loads. The data are representative of four independent experiments with similar results (**A**–**E**). (**A**,**B**,**D**) Significant differences (* *p* < 0.05; ** *p* < 0.01; *** *p* < 0.001) compared with rVehicle-treated BMDMs.

**Figure 6 ijms-21-08437-f006:**
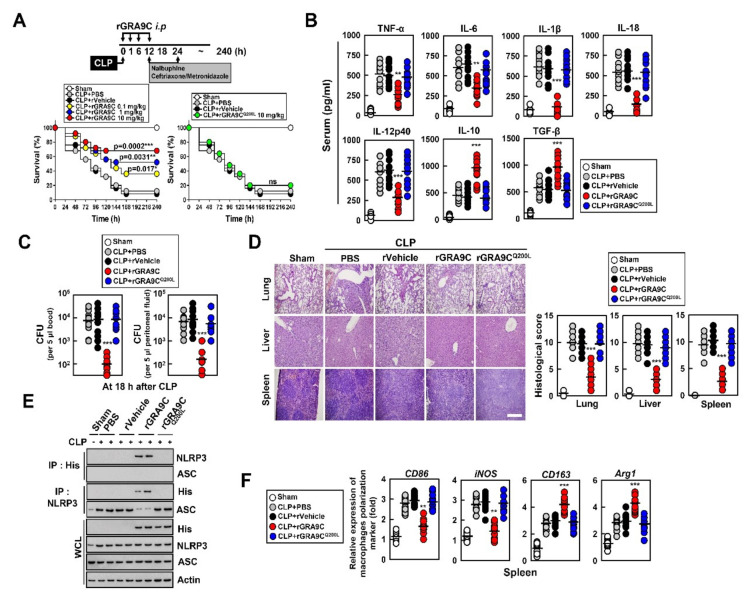
rGRA9C prevents mice from CLP-induced septic shock. (**A**) Schematic of the CLP model treated with PBS, rVehicle, rGRA9 at the indicated concentrations or rGRA9C^Q200L^ (upper). The survival of mice was monitored for 240 h; mortality was measured for *n* = 25 mice per group (lower). Statistical differences compared with the rVector-treated mice are indicated (log-rank test). The data are representative of two independent experiments with similar results. (**B**) Serum cytokine levels from 18 h after treatment of CLP mice with PBS, rVehicle, rGRA9C, or rGRA9C^Q200L^ per group (*n* = 10 mice per group). (**C**) The bacterial burden was evaluated 18 h after treatment of CLP mice with PBS, rVehicle, rGRA9C or rGRA9C^Q200L^ (*n* = 10 mice per group). (**D**) Representative hematoxylin and eosin (H&E) staining of the lung, liver and spleen (left) (*n* = 10 mice per group). Histopathology scores were obtained from H and E stained as described in methods (right) were determined at 30 h in CLP mice were treated with PBS, rVehicle, rGRA9, or rGRA9C^Q200L^. Scale bar, 500 μm. (**E**) Splenocytes were used for immunoprecipitation (IP) with αHis or αNLRP3, followed by immunoblotting (IB) with αNLRP3, αASC or αHis. WCLs were used for IB with αHis, αNLRP3, αASC or αActin. The data are representative of three independent experiments with similar results. (**F**) Splenocytes were used for quantitative real-time PCR with M1 (CD86 and iNOS) or M2 (CD163 or Arg1) macrophage polarization marker. Significant differences (* *p* < 0.05, ** *p* < 0.01; *** *p* < 0.001) compared with rVector-treated mice (**A**–**C**, **D** right, and **F**). Full-length images of the blots presented in the Appendix A.

**Figure 7 ijms-21-08437-f007:**
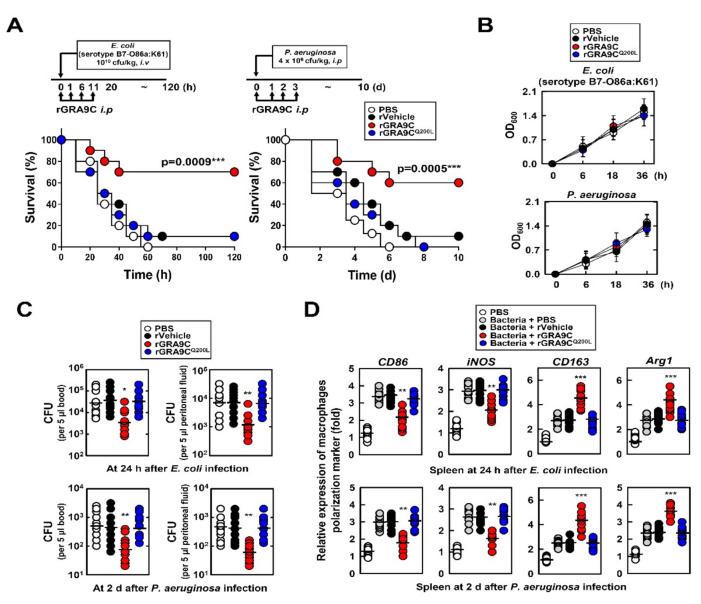
rGRA9C prevents mice from *E. coli* or *P. aeruginosa*-induced septic shock. (**A**) Schematic of the bacteria infection model treated with PBS, rVehicle, rGRA9, or rGRA9C^Q200L^ (upper). The survival of mice was monitored for 10 days; mortality was measured for *n* = 10 mice per group (lower). Statistical differences compared to the rVector-treated mice are indicated (log-rank test). The data are representative of two independent experiments with similar results. (**B**) Bacteria were cultured in LB broth contained in presence of PBS, rVector, rGRA9 or rGRA9C^Q200L^ (50 μg/mL) for the indicated times at 37 °C Measure the OD_600_ every 6 h. Data shown are the means ± SD of three experiments. (**C**) The bacterial burden was evaluated after 24 h or 2d in bacteria infected mice with PBS, rVehicle, rGRA9C or rGRA9C^Q200L^ (*n* = 10 mice per group). (**D**) Splenocytes were used for quantitative real-time PCR with M1 (CD86 and iNOS) or M2 (CD163 or Arg1) macrophage polarization marker. (**A**,**C**,**D**) Significant differences (* *p* < 0.05; ** *p* < 0.01; *** *p* < 0.001) compared with rVector-treated mice.

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
