# Peer review of "Toxoplasma gondii GRA9 Regulates the Activation of NLRP3 Inflammasome to Exert Anti-Septic Effects in Mice"

_ijms, 2020, doi:10.3390/ijms21228437_

Round 1

Reviewer 1 Report

The Authors presented a very interesting and well written manuscript. In my opinion, the methods and the whole study was well designed and the results obtained may have a great impact of the futher therapy in patients with sepsis. I have no suggestions and critical indications regarding the manuscript.

My only question is whether mice at similar body weight (years old) were used in case of CLP-induced sepsis and for cell culture, respectively?

Also, some minor english mistakes can be found

Author Response

Thanks for your kind comments. We used the similar body weight mice in CLP-induced sepsis and for cell culture. We have added age of mice in 4.1. Mice and cell culture section (line 400). Also, we have progressed additional english editing by Enago.

Reviewer 2 Report

Although the findings that Toxoplasma gondii GRA9 played a novel role in regulating NLRP3 inflammasome activation and macrophage polarization in the host are very much interesting, a number of points need clarifying and certain statements require further justification. These are given below.
  1. Concerning animal experiments, the authors described, “All animal experimental procedure were reviewed and approved by the Institutional Animal Care and Use Committee of Hanyang University (protocol 2018-0086)”. The authors should provide approval date.
2.        Fig. 4A (Coomassie blue staining) should be replaced by a much clearer figure. There is a large black band like photo stains.
3.        In Figure 5B, what is “1.0” should be explained. The Y-axis label is “Relative expression of macrophages polarization marker (fold)”. However, what is “1.0” was not explained. In contrast, the relative values of control (PBS/Sham) in Fig. 6F and 7D were “1.0”.
4.        In Fig. 6A, the maximum of X-axis was “240 h” although the authors described, “the survival of mice was monitored for 20 days” in the corresponding legend. Please make rational.
5.        The figure captions in Fig. 6C, 6D, 6E were too small. They should be much larger.
6.        The X- and Y-axis and their labels should be replaced to clearer letters.
Line 454: “ËšC” should be changed to “ËšC”.

Author Response

  1. Concerning animal experiments, the authors described, “All animal experimental procedure were reviewed and approved by the Institutional Animal Care and Use Committee of Hanyang University (protocol 2018-0086)”. The authors should provide approval date.

Thanks for your kind and insightful comments. We have added approval date in 4.6. CLP-induced sepsis and bacteria counts section (line 458).

  1. Fig. 4A (Coomassie blue staining) should be replaced by a much clearer figure. There is a large black band like photo stains.

Thanks for your kind and insightful comments. We have replaced Fig. 4A as a much clearer figure.

  1. In Figure 5B, what is “1.0” should be explained. The Y-axis label is “Relative expression of macrophages polarization marker (fold)”. However, what is “1.0” was not explained. In contrast, the relative values of control (PBS/Sham) in Fig. 6F and 7D were “1.0”.

Thanks for your kind and insightful comments. We have revised the data of Figure 5B for explaining “1.0”

  1. In Fig. 6A, the maximum of X-axis was “240 h” although the authors described, “the survival of mice was monitored for 20 days” in the corresponding legend. Please make rational.

We are sorry for our mistake. We have corrected line 262, as below,

The survival of mice was monitored for 240 h.

  1. The figure captions in Fig. 6C, 6D, 6E were too small. They should be much larger.

We are sorry for our mistake. We have enlarged the figure caption in Fig. 6C, 6D, 6E

  1. The X- and Y-axis and their labels should be replaced to clearer letters.

We are sorry for our mistake. We have edited X- and Y-axis labels in the figures to replaced clearer letters between Fig. 1 to Fig. 7.

Line 454: “ËšC” should be changed to “ËšC”.

We are sorry for our mistake. We revised the wrong word in line 454 (ºC -> ℃).

Round 2

Reviewer 2 Report

The authors revised the manuscript completely. It is a clear manuscript.